# Colossal flexoresistance in dielectrics

Sung Min Park[1,2], Bo Wang [3], Tula Paudel [4], Se Young Park[1,2,5], Saikat Das[1,2], Jeong Rae Kim[1,2], Eun Kyo Ko[1,2], Han Gyeol Lee[1,2], Nahee Park[6], Lingling Tao[4], Dongseok Suh [6], Evgeny Y. Tsymbal [4], Long-Qing Chen[3], Tae Won Noh [1,2✉] & Daesu Lee [7,8✉]

Dielectrics have long been considered as unsuitable for pure electrical switches; under weak electric fields, they show extremely low conductivity, whereas under strong fields, they suffer from irreversible damage. Here, we show that flexoelectricity enables damage-free exposure of dielectrics to strong electric fields, leading to reversible switching between electrical states —insulating and conducting. Applying strain gradients with an atomic force microscope tip polarizes an ultrathin film of an archetypal dielectric $SrTiO_3$ via flexoelectricity, which in turn generates non-destructive, strong electrostatic fields. When the applied strain gradient exceeds a certain value, $SrTiO_3$ suddenly becomes highly conductive, yielding at least around a $10^8$-fold decrease in room-temperature resistivity. We explain this phenomenon, which we call the colossal flexoresistance, based on the abrupt increase in the tunneling conductance of ultrathin $SrTiO_3$ under strain gradients. Our work extends the scope of electrical control in solids, and inspires further exploration of dielectric responses to strong electromechanical fields.

[1] Center for Correlated Electron Systems, Institute for Basic Science (IBS), Seoul 08826, Korea. [2] Department of Physics and Astronomy, Seoul National University, Seoul 08826, Korea. [3] Department of Materials Science and Engineering, The Pennsylvania State University, University Park, PA 16802, USA. [4] Department of Physics and Astronomy & Nebraska Center for Materials and Nanoscience, University of Nebraska, Lincoln, NE 68588, USA. [5] Department of Physics, Soongsil University, Seoul 07027, Korea. [6] Department of Energy Science, Sungkyunkwan University, Suwon 16419, Korea. [7] Department of Physics, Pohang University of Science and Technology (POSTECH), Pohang 37673, Korea. [8] Asia Pacific Center for Theoretical Physics, Pohang 37673, Korea. ✉email: twnoh@snu.ac.kr; dlee1@postech.ac.kr

Controlling electron dynamics in solids has opened avenues for fascinating physical phenomena[1–3] and has formed the basis of electronic applications. In semiconductors with a relatively small but nonzero bandgap, applying moderate electric fields could switch their electrical state, i.e., from insulator to conductor, which makes them a building block for contemporary digital electronics. In dielectrics with a large bandgap, controlling their electrical states is quite complicated, as it usually involves a combination of intrinsic and extrinsic processes. Zener[4] predicted that strong electric fields ($\geq 10^9$ V m$^{-1}$) could intrinsically lead to electrical breakdown in dielectrics through tunneling processes across the valence and conduction bands. As this dielectric breakdown naturally guarantees the largest and fastest electrical response, recent works have aimed to realize it by applying strong femtosecond fields[1,2]. Under strong static fields, however, the dielectric breakdown has been unavoidably subject to extrinsic effects[5,6], such as Joule heating and irreversible damage. This situation complicates our understanding of the intrinsic mechanism of dielectric breakdown and limits device application.

Here, we demonstrate that electrical states in dielectrics can be controlled by means of depolarization field induced by flexoelectric polarization. By applying the strain gradients from a conductive scanning probe tip, we simultaneously polarize and measure the current across the film. Above the certain critical strain gradients, the current–voltage (I–V) characteristic changes from tunneling-like to linear, which indicates the change of the electrical state from insulating to conducting. We explain this phenomena with a modulation of band structure due to the electrostatic field induced by flexoelectricity.

## Results

**Concept of flexoelectric control of electrical states in dielectrics.** To achieve intrinsic, static control of electrical states in dielectrics, we could utilize a non-destructive electrostatic field developed in ultrathin polar materials (Fig. 1a). When a polar material is sufficiently thin but still maintains polarization $P$, a depolarization field $E_{dep}$ arises from the unscreened bound charges on its surface[7,8]:

$$E_{dep} = -\frac{P - \sigma_S}{\varepsilon}, \qquad (1)$$

where $\sigma_S$ is the screening charge (e.g., by adjacent metal electrodes) and $\varepsilon$ is the dielectric permittivity of the polar material. In the ultrathin limit, $\sigma_S$ tends to zero[8] and $E_{dep}$ becomes increasingly saturated at $E_{dep} = -P/\varepsilon$, largely modifying the band structure (Fig. 1a). In particular, when the polarization exceeds a certain threshold, both the conduction band minimum and valence band maximum could cross the Fermi level, as confirmed in our first-principles calculation (Supplementary Fig. 1). In such a case, the tunnel-barrier width of ultrathin dielectrics would abruptly decrease, whereas the tunnel-barrier height remains fixed to the bandgap $\Delta_{bg}$ (Fig. 1a and Supplementary Fig. 2). This would result in a significant enhancement of tunneling conductance across ultrathin dielectrics, leading to a colossal decrease in electrical resistance, as predicted in our Wentzel–Kramers–Brillouin (WKB) simulation (Fig. 1b). Therefore, it would be of great interest to explore tunnel transport across a highly polarized ultrathin dielectric.

To this end, we can induce and stabilize large polarization in an ultrathin dielectric via flexoelectricity[9–20]. All dielectric materials polarize in response to strain gradients, as follows:

$$P = \varepsilon \cdot f_{eff} \cdot \frac{\partial u}{\partial x}, \qquad (2)$$

where $\partial u / \partial x$ and $f_{eff}$ are the strain gradient and effective flexocoupling coefficient, respectively. Applying loading forces through an atomic force microscope (AFM) tip (Fig. 2a) generates strain gradients as large as $10^7$ m$^{-1}$ in ultrathin dielectrics[13,17–19]. Such giant strain gradient could then induce flexoelectric polarization, up to a few 0.1 Cm$^{-2}$ (ref. [19]), much larger than the polarization values typically attainable in ultrathin ferroelectrics[21,22].

**Colossal flexoresistance in an archetypal dielectric SrTiO₃.** We choose SrTiO₃ (STO) as a model dielectric system, as it shows enhanced flexocoupling strength at the nanoscale[19], as well as a reasonably large bandgap of 3.2 eV. Importantly, furthermore, its conductivity responds negligibly to the applied strain itself (Supplementary Fig. 3)[23,24], thereby maximizing the contribution from strain gradient-induced flexoelectricity. We prepare 10-unit-cell-thick (i.e., 3.9-nm thick) stoichiometric STO films on a (001)-oriented STO single crystal substrate, with a conductive SrRuO₃ buffer layer (Supplementary Figs. 4 and 5). The stoichiometric homoepitaxial STO should remain paraelectric down to 0 K

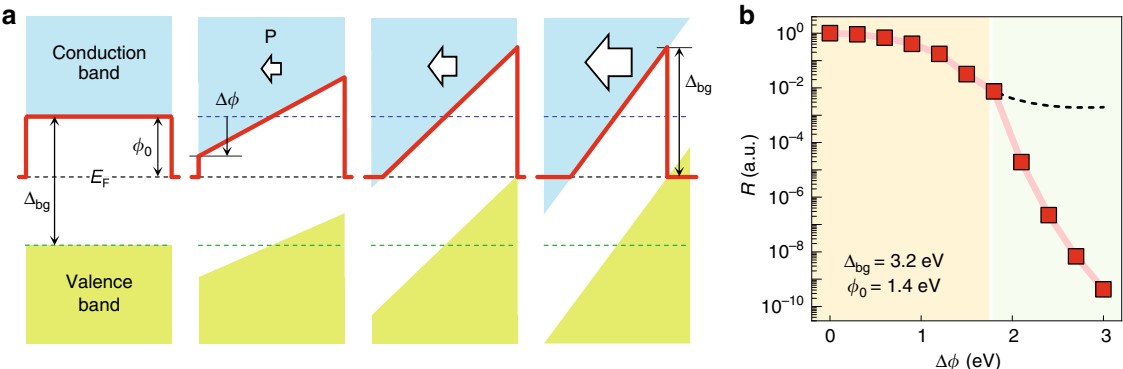

**Fig. 1 Colossal decrease of resistivity in highly polarized ultrathin dielectrics. a** Schematic diagram of the potential energy profiles across SrTiO₃ (STO) with increasing flexoelectric polarization (**P**; white arrow). Red solid lines and black dashed lines indicate the effective tunnel barrier and Fermi level, respectively. Blue and green dashed lines indicate the conduction band minimum and valence band maximum for **P** = 0, respectively. **b** Resistance as a function of $\Delta\varphi$, obtained by calculating tunneling conductance through a Wentzel–Kramers–Brillouin (WKB) approximation. We normalize the resistance by the value at $\Delta\varphi = 0$, and assume the bandgap $\Delta_{bg}$, original barrier height $\varphi_O$, and original barrier width $d_O$ to be 3.2 eV, 1.4 eV, and 3.9 nm, respectively. At $\Delta\varphi = 1.8$ eV, the valence band maximum crosses the Fermi level, which causes an abrupt reduction in the resistance. The black dashed line indicates the result obtained by neglecting the valence band contribution. Source data are provided as a Source data file.

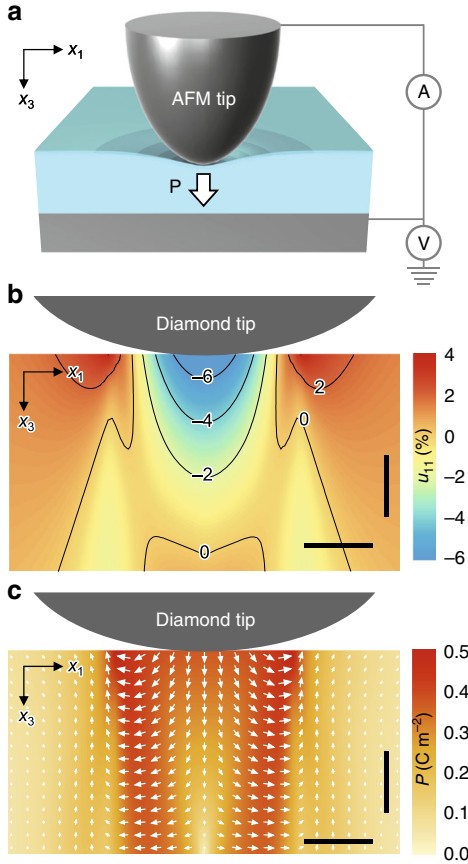

**Fig. 2 Mechanically induced large polarization in ultrathin dielectrics.**
**a** Schematic diagram of the experimental setup, illustrating the flexoelectric polarization (white arrow) generated by the atomic force microscope (AFM) tip pressing the surface of ultrathin dielectrics. While generating large strain gradients, we simultaneously measure the tunneling currents across the flexoelectrically polarized STO. **b, c** Phase-field simulations for the transverse strain $u_{11}$ (**b**) and corresponding polarization distribution (**c**) in ultrathin STO under a representative tip loading force of 15 μN over a circular area ~13 nm in radius. Vertical and horizontal scale bars represent 1 nm and 10 nm, respectively. Source data are provided as a Source data file.

(ref. [25]); however, under an AFM-tip loading force, it can become highly polarized via flexoelectricity[19].

We then use contact mechanics analysis to simulate strain gradients and associated flexoelectric polarization in ultrathin STO under an AFM-tip loading force (see Methods). For the simulation, we adopt a diamond tip and assume a tip radius of curvature ($r_{tip}$) of 100 nm. Note that the actual contact radius is estimated to be around 13 nm for the case of a 15 μN tip loading force, which is much smaller than the tip radius $r_{tip}$. Figure 2b shows a simulated profile of transverse strain $u_{11}$ under a representative tip loading force of 15 μN, revealing the large inhomogeneity of $u_{11}$. The resulting transverse strain gradients $\partial u_t/\partial x_3$ (i.e., $=\partial u_{11}/\partial x_3 + \partial u_{22}/\partial x_3$) are as huge as a few $10^7$ m$^{-1}$ (Supplementary Fig. 6); this giant strain gradients are attributable to AFM-tip-induced downward bending at the nanoscale. Our simulation also finds that those strain gradients induce large polarization in ultrathin STO via flexoelectricity, reaching up to 0.18 Cm$^{-2}$ on average (Fig. 2c). When neglecting flexoelectricity, our simulation does not produce any polarization, confirming the flexoelectric nature of the induced polarization.

When such a large polarization remains preserved in an ultrathin dielectric, it could significantly modify the band

structure of the dielectric, as predicted in Fig. 1. We estimate the threshold polarization $P_{th}$ in ultrathin STO, above which both the conduction band minimum and valence band maximum cross the Fermi level (Fig. 1a):

$$|P_{th}| \approx \varepsilon \cdot E_{dep,th} = \varepsilon \cdot \frac{\Delta_{bg}}{e \cdot t}, \qquad (3)$$

where $\Delta_{bg}$ and $t$ are the bandgap and thickness of the STO layer, respectively, $e$ is the electronic charge, and $E_{dep,th}$ is the threshold $E_{dep}$. Given that $\varepsilon \sim 20\varepsilon_0$ of strained STO (Supplementary Fig. 7), $\Delta_{bg} = 3.2$ eV and $t = 3.9$ nm, Eq. (3) yields $P_{th} = 0.15$ Cm$^{-2}$, comparable to the value obtained in our simulation (Fig. 2c). At a certain AFM-tip loading force, therefore, the induced flexoelectric polarization could give rise to an abrupt increase in tunneling currents across ultrathin STO.

Motivated by this, we use a conductive AFM tip to apply loading forces while simultaneously measuring the tunneling current (Fig. 2a). Conforming to the simulation condition, we use a diamond-coated tip with $r_{tip} = 100$ nm (Supplementary Fig. 8), which also can withstand much higher loading forces than other conductive tips (e.g., PtIr-coated tips). Figure 3a shows $I$–$V$ curves measured at room temperature for a few representative loading forces (see Supplementary Fig. 9 for the entire set). At small applied forces (up to 7 μN), the measured current remains close to the noise level (a few pA). At intermediate applied forces (ranging from 7 to 13 μN), the $I$–$V$ curves exhibit typical tunneling characteristics, and the current level increases gradually with the applied force. These results are ascribable to systematic modification of tunnel-barrier profiles under AFM-tip loading forces, consistent with our previous work[19]. Interestingly, when the applied forces exceed a threshold value (~15 μN), the $I$–$V$ curves suddenly become linear in shape—characteristic of a highly conducting state. This highlights that the electrical state of ultrathin STO is switchable between highly insulating and conducting states, via purely mechanical means.

Importantly, this electrical-state switching in a large-bandgap dielectric naturally leads to an extremely large change in the electrical resistivity. During electrical-state switching, the effective resistivity of STO exhibits a colossal change with around eight orders-of-magnitude difference (Fig. 3b; see also Methods). Due to the detection limit of our equipment, we may underestimate the resistivity of the insulating state, i.e., $10^7$ Ω cm, compared with the bulk STO resistivity of over $10^9$ Ω cm; thus, the actual ratio of resistivity change could be larger than the estimated value. Given that we estimate the effective resistivity by taking into account the loading force dependence of the tip–STO contact area and STO thickness, we exclude any geometric anomaly as an origin for the observed effect. When we normalize the measured effect by applied pressures (i.e., loading forces divided by the tip–STO contact area), the relative increase in conductivity turns out to be as large as $10^{-3}$–$10^{-2}$ Pa$^{-1}$. Compared with other pressure-induced effects, such as piezoresistance (at most, $10^{-7}$ Pa$^{-1}$)[26,27], this effect not only shows several orders-of-magnitude enhancement, but also implies a distinctly new mechanism.

**Excluding other origins.** Before addressing how flexoelectricity could explain our results, we rule out other possible origins of the phenomenon. We first exclude any involvement of an electro-chemical process. The AFM-tip-induced mechanical force does not cause any permanent surface damage to the STO film (Fig. 4a, b), and the colossal control of resistance is reproducible, as proven by repeated exertion/withdrawal of the loading force (Fig. 4c). We also reproduce the same result even using a graphene top electrode (Supplementary Fig. 10). This again excludes any electrochemical interaction of STO with experimental environments, such as the AFM tip or ambient atmosphere, as graphene is

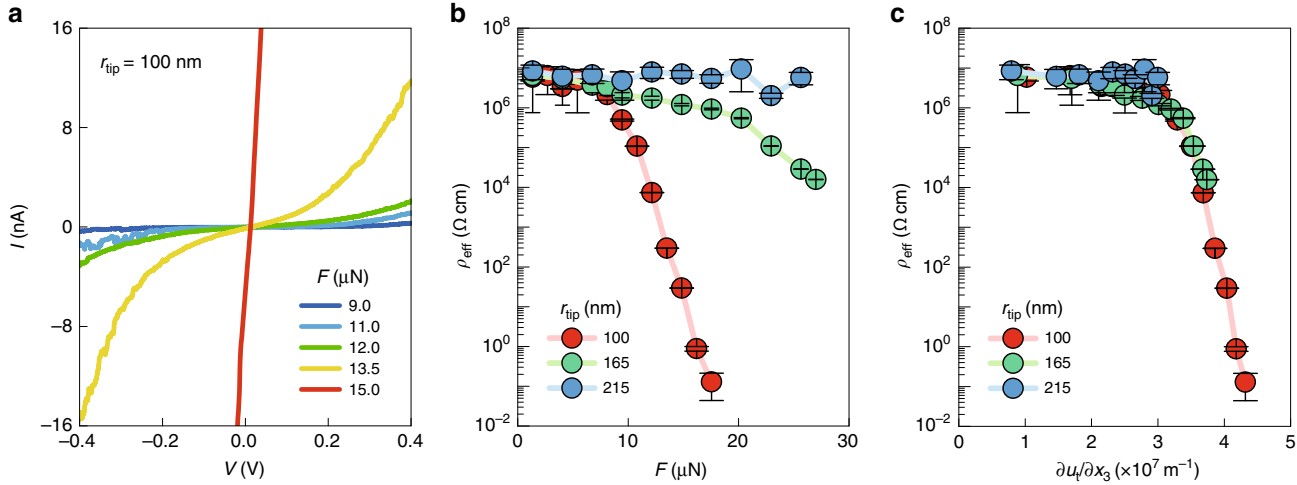

**Fig. 3 Colossal flexoresistance effect in ultrathin STO. a** Current–voltage (I–V) curves obtained by conductive AFM measurements in a 10-unit cell-thick STO film upon application of various tip loading forces F. Five representative curves are shown here. **b** Effective resistivity ($\rho_{eff}$) as a function of F. **c** $\rho_{eff}$ as a function of the AFM-tip-induced strain gradient $\partial u_t/\partial x_3$. Error bars denote standard deviations of the fitted resistivity. Source data are provided as a Source data file.

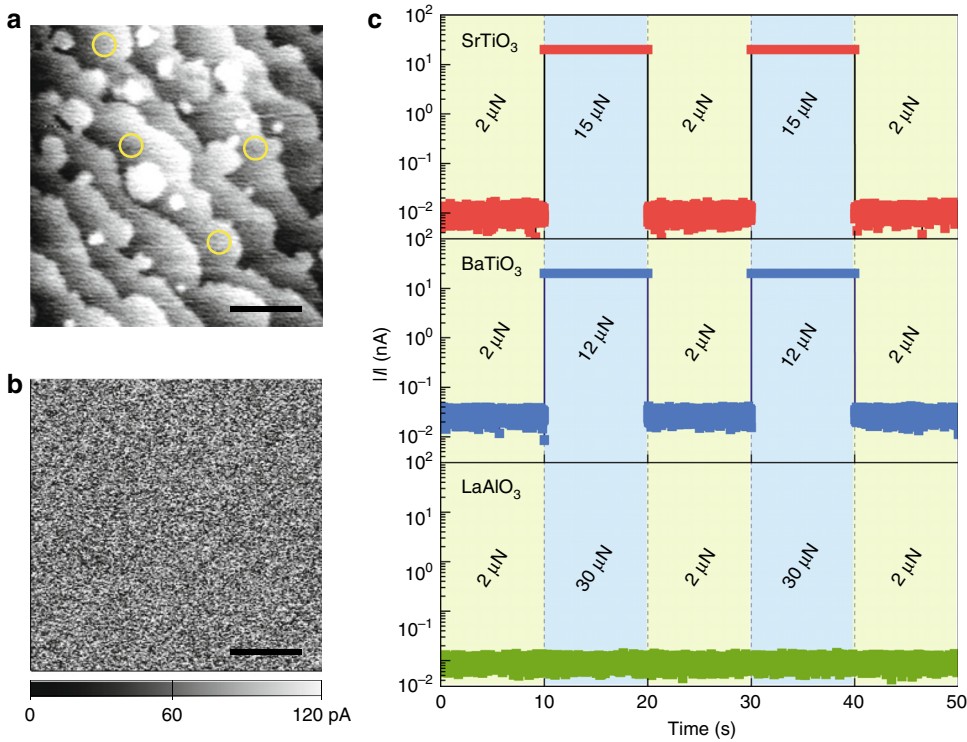

**Fig. 4 Reversible flexoresistance effect. a** Topographical image obtained after the experiment. Regions where the experiments were conducted are marked by yellow circles. **b** Current mapping image on the same area, recorded with a 1-V bias voltage under a constant tip's loading force of 2 μN, indicating that no current hotspot has been made after the experiment. **c** Current measured with a 0.1-V bias voltage under two representative loading forces in STO (red symbols), BaTiO₃ (blue symbols) and LaAlO₃ (green symbols). The lower threshold loading force (i.e., around 12 μN) for BaTiO₃ may originate from inherently stronger flexocoupling strength in BaTiO₃ (ref. [29]), compared with that in STO. During the measurements, we set the current limit (compliance) to 20 nA. Scale bars in (**a**) and (**b**) represent 2 μm. Source data are provided as a Source data file.

impermeable to all atoms and molecules. In addition, based on quantitative and qualitative evidences (Supplementary Figs. 11–13), we exclude an electrostatic interaction between the AFM tip and STO as the primary origin of the colossal resistivity change observed.

Furthermore, the AFM-tip-induced strain itself cannot largely change the resistivity of STO that does not have $d$ electrons. According to our theoretical analysis, the AFM-tip loading force generates compressive longitudinal strain $u_{33}$ up to around 0.15 in STO (Supplementary Fig. 6). Because the antibonding and bonding

states of Ti $3d$ and O $2p$ orbitals form the conduction and valence bands of STO, respectively, the compressive strain rather increases the bandgap of STO slightly (Supplementary Fig. 3)[23,24]; the effect of strain itself thus cannot explain the observed colossal decrease in STO resistivity, distinct from conventional piezoresistance effects[26,27]. For confirming this in our geometry, we repeat the experiments using AFM tips with different $r_{tip}$ values of ~165 and ~215 nm (Supplementary Fig. 8). Although these tips can generate longitudinal strain $u_{33}$, comparable to that by the tip with $r_{tip} = 100$ nm (Supplementary Fig. 6), the resulting resistivity changes are suppressed considerably (Fig. 3b and Supplementary Fig. 14). Therefore, these results suggest that the observed colossal decrease in resistivity could originate from AFM-tip-induced strain gradients, which modify the tunnel barrier via flexoelectricity.

**Strain-gradient-dependent resistivity change.** Figure 3c indeed highlights the close correlation between the resistivity change and strain gradients. Strikingly, all of the data obtained with the three tips collapse to a nearly single curve when plotting the resistivity as a function of $\partial u_t/\partial x_3$. This emphasizes the dominant contribution of $\partial u_t/\partial x_3$ to the observed colossal reduction of resistivity. As predicted in Fig. 1, when the strain gradient-induced flexoelectric polarization reaches a threshold value, both the conduction band minimum and valence band maximum cross the Fermi level. This band crossing is capable of not only enhancing the tunneling conductance across STO (Fig. 1b) but also promoting interband tunneling between the STO valence and conduction bands, causing Zener breakdown[3,4,28]. Equations (2) and (3) estimate the threshold $\partial u_t/\partial x_3$ required for the band crossing to be around $3 \times 10^7$ m$^{-1}$ for 10-unit-cell-thick STO, using $\Delta_{bg} = 3.2$ eV, $t = 3.9$ nm, and $f_{eff} = 25$ V (ref. [19]). This agrees quantitatively with our experimental results (Fig. 3c), in which the colossal decrease of resistivity begins at $\partial u_t/\partial x_3 \sim 3.5 \times 10^7$ m$^{-1}$. Taken together, our experimental and theoretical results consistently evidence that flexoelectric polarization-induced band crossing could explain the colossal reduction of resistivity, which we call the colossal flexoresistance.

**Colossal flexoresistance in various dielectrics.** As such, it would be interesting to explore colossal flexoresistance in other dielectrics. As flexoelectricity is a universal phenomenon in all dielectrics, colossal flexoresistance could, in principle, be universal as well. Each dielectric, however, would require different threshold loading forces (i.e., threshold strain gradients) for colossal flexoresistance, depending on the inherent flexocoupling strength, bandgap, and so on. We repeat the experiments using BaTiO$_3$, CaTiO$_3$, and LaAlO$_3$ of similar thicknesses (i.e., 10-unit-cell thick). For BaTiO$_3$ and CaTiO$_3$, we observe the same electrical-state switching (Fig. 4c and Supplementary Fig. 15), but at lower and higher threshold loading forces, respectively. The lower (or higher) threshold loading force for BaTiO$_3$ (or CaTiO$_3$) may originate from inherently stronger (or weaker) flexocoupling strength[29] and/or smaller (or larger) bandgap, compared with those in STO. For LaAlO$_3$, contrarily, we does not observe any noticeable resistance change up to the maximum AFM-tip loading force (Fig. 4c). LaAlO$_3$ may have a much weaker flexocoupling strength due to its small Born effective charge[30]; additionally, its large bandgap (i.e., $\Delta_{bg} = 5.5$ eV) also requires a large threshold polarization in Eq. (3). These conditions may make the threshold strain gradient in LaAlO$_3$ too large to be achievable in our experimental geometry.

## Discussion

The colossal flexoresistance effect described here overcomes a long-standing dilemma: the electrical-state switching in dielectrics requires strong fields, but when applied by strong static fields,

dielectrics inevitably suffer from irreversible damage. Utilizing universal flexoelectricity, we develop a general approach to apply non-destructive, strong electrostatic fields in various insulating systems, such as the Mott insulator[3]. Our approach will open up new avenues for unconventional quantum phenomena under strong static fields and device applications, such as the flexoelectronic transistor and mechanical sensor.

## Methods

**First-principles calculations.** The atomic and electronic structures of the system were obtained using density functional theory (DFT) implemented in the Vienna ab initio simulation package (VASP)[31,32]. The projected augmented plane wave (PAW) method was used to approximate the electron–ion potential[33]. The exchange and correlation potentials were calculated using the local spin density approximation (LSDA). In the calculations, we employed a kinetic energy cutoff of 340 eV for PAW expansion, and a $6 \times 6 \times 1$ grid of **k** points[34] for Brillouin zone integration. The in-plane lattice constant was that of relaxed bulk STO ($a = 3.86$ Å); the $c/a$ ratio and internal atomic coordinates were relaxed until the Hellman–Feynman force on each atom fell below $|0.01|$ eV Å$^{-1}$.

To understand the effect of electronic polarization on the interfacial electronic structure, we constructed a SrRuO$_3$/STO bilayer with five unit cells of SrRuO$_3$ and nine layers of STO, part of which is shown in Supplementary Fig. 1a. The sub-interfacial layers of the completely relaxed paraelectric phase of STO on SrRuO$_3$ are insulating, and the Fermi level lies in the gap between the conduction band minima and valence band maxima. However, when STO is highly polarized, the induced electrostatic field largely bends bands; thus, both the conduction band minimum and valence band maximum of sub-interfacial STO layers could cross the Fermi level, as shown in the LDOS plot (Supplementary Fig. 1b). We plotted Supplementary Fig. 1b with frozen uniform displacement of the Ti atom by 0.18 and 0.54 Å. Note that polarized tetragonal STO has higher energy than paraelectric cubic STO, but can be stabilized under non-equilibrium strain conditions[35]. This band profile clearly supports the experimental finding that the band crossing of STO conduction and valence bands could lead to a colossal decrease in the electrical resistivity.

The dielectric constant was calculated using density functional perturbation theory[36–38]. Supplementary Fig. 7 represents the calculated total $zz$ component of the total dielectric constant (i.e., $\varepsilon_{zz}$), which includes both ionic and electronic contributions, as a function of strain $u$. The strain was measured with respect to the DFT equilibrium lattice of 3.86 Å.

In order to investigate the change in the bandgap of STO in the presence of the strain, we have used the hybrid functional (HSE06)[39] implemented in the VASP package (Supplementary Fig. 3)[31,32]. We have used a 5-atom unit cell to simulate unstrained cubic and strained tetragonal structures in which an $8 \times 8 \times 8$ **k**-point grid with energy cutoff of 600 eV are used. Convergence is reached if the consecutive energy difference is within 0.01 meV for electronic iterations and 0.1 meV for ionic relaxations. The calculated lattice constant of the cubic structure by full structure relaxation is 3.897 Å with a bandgap of 3.3 eV, in good agreement with experimental data[40,41]. For the strained tetragonal unit cell, a unit cell with compressive strain of 2 and 10% for $a$ and $c$ lattice constants ($a = 3.819$ Å, $c = 3.507$ Å) with respect to the relaxed cubic structure is considered based on the strain profile simulation. The calculated bandgap of the tetragonal structure is around 3.6 eV, slightly larger than that of the cubic structure.

**Wentzel–Kramers–Brillouin simulation.** Using the one-dimensional WKB approximation, we can simply describe the tunneling current density for a low $T$ and small $V$, as follows:

$$j(V) = \frac{2e}{h} \int_{-\infty}^{\infty} T(E) \times [f(E) - f(E - eV)]dE$$
$$\cong \frac{2e}{h} \int_{-\infty}^{\infty} \exp\left(-\frac{4\pi}{h}\int_0^d \sqrt{2m(U(x)-E)}dx\right) \times [f(E) - f(E-eV)]dE \quad (4)$$
$$\cong \frac{2e}{h}\exp\left(-\frac{4\pi}{h}\int_0^d \sqrt{2m(U(x)-E_F)}dx\right) \times eV,$$

where $T(E)$, $f(E)$, $U(x)$, and $m$ represent the transmission probability, Fermi–Dirac distribution, tunnel-barrier profile, and free electron mass, respectively. Using Eq. (4), we obtain the tunneling current density for a trapezoidal barrier profile (Supplementary Fig. 2) as follows[19]:

$$j(+V) = \frac{2e}{h}\exp\left(-\frac{4\pi}{h}\int_0^{d_0}\sqrt{2m\left\{\frac{\varphi_2 - \varphi_1 + eV}{d_0}(x - d_0) + \varphi_2\right\}}dx\right) \times eV$$
$$= \frac{2e}{h}\exp\left(-\frac{8\pi\sqrt{2m}}{3h}\cdot d_0 \cdot \frac{(\varphi_2)^{1.5} - (\varphi_1 - eV)^{1.5}}{\varphi_2 - \varphi_1 + eV}\right) \times eV, \quad (5)$$

$$j(-V) = -\frac{2e}{h}\exp\left(-\frac{4\pi}{h}\int_0^{d_0}\sqrt{2m\left\{\frac{\varphi_2-\varphi_1-eV}{d_0}(x-d_0)+\varphi_2-eV\right\}}dx\right)\times eV$$
$$= -\frac{2e}{h}\exp\left(-\frac{8\pi\sqrt{2m}}{3h}\cdot d_0 \cdot\frac{(\varphi_2-eV)^{1.5}-(\varphi_1)^{1.5}}{\varphi_2-\varphi_1-eV}\right)\times eV,$$

(6)

where $\phi_2$ and $\phi_1$ are the barrier heights of the right and left sides of the trapezoidal barrier, respectively, i.e., $\phi_1 = \phi_0 + \Delta\phi$ and $\phi_1 = \phi_0 - \Delta\phi$. Using Eq. (4), we can also obtain the tunneling current density for a triangular barrier profile (Supplementary Fig. 2) as follows[19]:

$$j(+V) = \frac{2e}{h}\exp\left(-\frac{4\pi}{h}\int_0^{d'}\sqrt{2m\left\{\frac{\varphi}{d'}(x-d')+\varphi\right\}}dx\right)\times eV$$
$$= \frac{2e}{h}\exp\left(-\frac{8\pi\sqrt{2m}}{3h}\cdot d\cdot\frac{\varphi^{1.5}}{\varphi+eV}\right)\times eV,$$

(7)

$$j(-V) = -\frac{2e}{h}\exp\left(-\frac{4\pi}{h}\int_0^{d}\sqrt{2m\left\{\frac{\varphi-eV}{d}(x-d)+\varphi-eV\right\}}dx\right)\times eV$$
$$= -\frac{2e}{h}\exp\left(-\frac{8\pi\sqrt{2m}}{3h}\cdot d\cdot(\varphi-eV)^{0.5}\right)\times eV,$$

(8)

where $\phi$ and $d$ indicate the barrier height and width of the triangular barrier, and $d' = d\cdot\phi/(\phi+eV)$. Importantly, depending on whether we consider the contribution of the STO valence band, $\phi$ and $d$ have a different dependence on $\Delta\phi$. When neglecting the STO valence band, $\phi = \phi_0 + \Delta\phi$ and $d = d_0\cdot[(\phi_0+\Delta\phi)/2\Delta\phi]$ (Supplementary Fig. 2a); this indicates that although the increased $\Delta\phi$ reduces the barrier width $d$, it also increases the barrier height $\phi$, such that the overall tunneling conductance cannot become largely enhanced (as shown in Fig. 1b; black dashed line). In striking contrast, when considering the STO valence band, $\phi$ is fixed to $\Delta_{bg}$ and $d = d_0\cdot\Delta_{bg}/2\Delta\phi$ (Supplementary Fig. 2b), which could lead to colossal enhancement of the tunneling conductance with increasing $\Delta\phi$ (Fig. 1b).

**Sample fabrication.** Ultrathin STO, BaTiO$_3$, CaTiO$_3$, and LaAlO$_3$ films were grown by pulsed laser deposition, using a KrF excimer laser ($\lambda = 248$ nm). STO, BaTiO$_3$, and LaAlO$_3$ films were grown on bottom electrodes of epitaxial 20-nm-thick SrRuO$_3$, prepared on TiO$_2$-terminated and (100)-oriented STO substrates. CaTiO$_3$ films were grown on LaAlO$_3$ substrates buffered by LaNiO$_3$ conducting electrode. The growth patterns and thickness were monitored by in situ reflection high-energy electron diffraction (RHEED; Supplementary Fig. 4). We deposited SrRuO$_3$ and STO thin films at 700 °C under oxygen partial pressure of 100 and 7 mTorr, respectively. After deposition, films were annealed at 475 °C for 1 h in oxygen at ambient pressure and subsequently cooled to room temperature at 50 °C min$^{-1}$. X-ray diffraction reciprocal space mapping was performed to ensure that the STO film was strain-free (Supplementary Fig. 5). Piezoresponse force microscopy found that as-grown BaTiO$_3$ films had downward self-polarization (Supplementary Fig. 11).

**Simulation of strain profile.** The strain distribution in ultrathin STO film pressed with an AFM tip is obtained by solving the elastic equilibrium equation in a 3D thin film/substrate system with appropriate boundary conditions. The detailed procedure is described in previous works[42]. Here, we discretized three-dimensional space into $100\times100\times500$ grid points and applied periodic boundary conditions along the $x_1$ and $x_2$ axes. The grid spacing was $\Delta x_1 = \Delta x_2 = 0.5$ nm and $\Delta x_3 = 0.1$ nm. Along the $x_3$ direction, 35 layers were used to mimic the film; the relaxation depth of the substrate featured 350 layers to ensure that the displacement was negligibly small. To estimate the surface stress distribution that developed with AFM-tip pressing, we adopted the closed-form solution derived by Wang et al. for indentation responses in a piezoelectric thin film in the ultrathin-film limit[43]. This contact mechanics model, comparing to the classical Hertz model for a non-piezoelectric, semi-infinite space[44], considers not only the finite size of the film but also the coupled nature of the indentation problem of an electromechanically active sample. Therefore, it is more appropriate to describe the surface stress caused by nano-indentation in ultrathin STO films in the present work.

We considered an STO thin film of thickness $h_f$, with the top surface in contact with an AFM tip and the bottom interface coherently constrained by the substrate. We assumed a conductive, rigid, spherical indenter with a tip radius $r_{tip} = 100$ nm and a mechanical force $F$ ranging from 1 to 25 μN. At the top surface, the normal stress distribution (as a function of the distance from the contact center) is described as follows:

$$\sigma_{33}^{tip}(r) = \begin{cases} -c_{33}p\frac{(a^2-r^2)}{2h_{def}}, & r \le a \\ 0, & r \ge a \end{cases},$$

(9)

where $a$ is the contact radius $\left(a = \sqrt{2R(h_{ind}+\frac{e_{33}}{c_{33}}\phi_0)}\right)$, $h_{ind}$ is the indentation depth $\left(h_{ind} = \sqrt{\frac{Fh_{def}}{c_{33}\pi R}}-\frac{e_{33}}{c_{33}}\phi_0\right)$, $h_{def}$ is the deformation depth $h_{def} = h_f$, $c_{33}$ is the

elastic stiffness of the STO film ($c_{33} = 336$ GPa), $e_{33}$ is the piezoelectric coefficient of the STO film ($e_{33} = 0$ Cm$^{-2}$), and $\phi_0$ is the applied bias ($\phi_0 = 0$ V). At the film–substrate interface, the displacement is continuous for coherency and is assumed to relax to zero within a depth of $h_s$ into the substrate (i.e., $\eta_i|_{x_3=-h_s} = 0$). The clamping effect of the STO substrate is considered to render the average strain zero at each layer of the film (i.e., $\overline{u_{11}} = \overline{u_{22}} = 0$ and $\overline{u_{12}} = 0$). Finally, the boundary value problem of elastic equilibrium, assuming no body force, is given by

$$\begin{cases} \sigma_{ij,j} = 0 \\ \sigma_{33}|_{x_3=h_f} = \sigma_{33}^{tip}, \sigma_{31}|_{x_3=h_f} = \sigma_{32}|_{x_3=h_f} = 0, \\ \eta_i|_{x_3=-h_s} = 0 \end{cases}$$

(10)

where stress is related to strain via $\sigma_{ij} = c_{ijkl}e_{kl} = c_{ijkl}(u_{kl}-u_{kl}^0)$. The eigenstrain $u_{ij}^0$ is derived from strain-order parameter couplings of STO through $u_{ij}^0 = Q_{ijkl}P_kP_l + \lambda_{ijkl}q_kq_l$, where $Q_{ijkl}$ and $\lambda_{ijkl}$ are the electrostrictive and rotostrictive tensors, respectively. The electrostrictive and rotostrictive coupling coefficients of STO were adapted from ref. [45].

**Simulation of the polarization profile.** The polarization distribution under the mechanical load by an AFM tip can be calculated by self-consistent phase-field modeling[46]. The temporal evolution of the polarization field $\mathbf{P}(\mathbf{x},t)$ is governed by the time-dependent Ginzburg–Landau equation, i.e., $\partial\mathbf{P}/\partial t = -L(\delta F(\mathbf{P})/\delta\mathbf{P})$, where $L$ is the kinetic coefficient and the total free energy $F$ can be expressed as[46]

$$F = \int \left(f_{bulk} + f_{elastic} + f_{electric} + f_{gradient} + f_{flexo}\right)dV$$
$$= \int \left[\alpha_{ij}P_iP_j + \alpha_{ijkl}P_iP_jP_kP_l + \beta_{ij}\theta_i\theta_j + \beta_{ijkl}\theta_i\theta_j\theta_k\theta_l + t_{ijkl}P_iP_j\theta_k\theta_l + \frac{1}{2}g_{ijkl}\frac{\partial P_i}{\partial x_j}\frac{\partial P_k}{\partial x_l}\right.$$
$$\left. +\frac{1}{2}k_{ijkl}\frac{\partial\theta_i}{\partial x_j}\frac{\partial\theta_k}{\partial x_l} + \frac{1}{2}c_{ijkl}(u_{ij}-u_{ij}^0)(u_{kl}-u_{kl}^0) - \frac{1}{2}E_iP_i + \frac{1}{2}f_{ijkl}\left(\frac{\partial P_k}{\partial x_l}\varepsilon_{ij}-\frac{\partial\varepsilon_{ij}}{\partial x_l}P_k\right)\right]dV.$$

(11)

The bulk Landau free energy $f_{bulk}$ consists of two sets of order parameters, i.e., the spontaneous polarization $\mathbf{P}$ and the antiferrodistortive order parameter $\mathbf{\theta}$, which represents the oxygen octahedral rotation angle of STO[45]. The flexoelectric contribution is considered as a Liftshitz invariant term as

$$f_{flexo} = \frac{1}{2}f_{ijkl}\left(\frac{\partial P_k}{\partial x_l}u_{ij} - \frac{\partial u_{ij}}{\partial x_l}P_k\right).$$

(12)

The eigenstrain tensor $\mathbf{u}^0$ in the elastic energy density is given by

$$u_{ij}^0 = Q_{ijkl}P_kP_l + \Lambda_{ijkl}\theta_k\theta_l - F_{ijkl}P_{k,l},$$

(13)

where the electrostrictive, rotostrictive, and converse flexoelectric couplings are considered via tensors $\mathbf{Q}$, $\mathbf{\Lambda}$, and $\mathbf{F}$. The coefficients used in constructing the total free energy $F$ of an STO single crystal are given in our previous works[45,47]. The transverse flexoelectric constant of STO estimated from experiments in the previous work was used ($f_{12} = 25$ V)[19]; the other two flexoelectric components were assumed to be zero (i.e., $f_{11} = f_{44} = 0$) for simplicity.

**Tunneling measurements.** The $I$–$V$ curves were obtained using an Asylum Research Cypher AFM (Oxford Instruments, Abingdon, UK) at room temperature under ambient conditions. Conducting diamond-coated metallic tips (DDESP-V2; BRUKER, Billerica, MA, USA) with nominal spring constants 80 Nm$^{-1}$ and a dual-gain ORCA module (Oxford Instruments) were used to measure currents. In order to estimate the $r_{tip}$ from the measured scanning electron microscopy (SEM) images (Supplementary Fig. 8a–c), we digitized the profile of the tip shape using a Java-based software (plot digitizer 2.6.8). The outline of the tip was tracked down with a scale of a pixel (~35 nm) in SEM images. Digitized data points were fitted with parabolic function $\Delta z = c_2(\Delta x)^2 + c_1(\Delta x) + c_0$ (Supplementary Fig. 8d–f), then converted into $r_{tip}$ as $r_{tip} = 1/|2c_2|$.

An electrical bias was applied through the conducting SrRuO$_3$ electrode; this was swiped from -0.5 to +0.5 V at a ramping rate of about 4 Vs$^{-1}$. During the measurements, we set the current limit (compliance) to 20 nA. The noise floor of the AFM system was a few pA. We measured the resistance $R$ from the linear slope of $I$–$V$ curves in the low-bias regime. We extracted the resistance of STO, i.e., $R_{STO}$, from the difference between the measured $R$ and the resistance of the bottom SrRuO$_3$ layer (i.e., ~70.4 kΩ; Supplementary Fig. 9i). We then estimated an effective resistivity ($\rho_{eff}$) of STO by considering the effective tip–STO contact radius ($a$) and the effective STO thickness ($t_{STO}$):

$$\rho_{eff} = R_{STO}\cdot\frac{\pi a^2}{t_{STO}},$$

(14)

where we obtained the values of $a$ and $t_{STO}$ from our theoretical contact mechanics analysis.

**Graphene.** For the graphene transfer onto the ultrathin BaTiO$_3$ film, we followed the so-called dry-transfer technique. A graphene monolayer was mechanically

exfoliated on a silicon wafer coated with poly(vinyl alcohol) (PVA), which is water-soluble, and poly(methyl methacrylate) (PMMA). After the selection of a proper graphene flake, the flake/PMMA layer was detached from the silicon substrate by immersion in hot deionized water. Then, the flake/PMMA layer floating on the water was transferred to a holder and was placed on the ultrathin $BaTiO_3$ film using a homemade micromanipulator after alignment under an optical microscope. At last, the PMMA was removed with acetone.

## Data availability
All relevant data presented in this paper are available from the authors upon reasonable request. The source data underlying Figs. 1–4 and Supplementary Figs. 5–11,13–15 are provided as a Source data file.

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

## Acknowledgements
This work was supported by the Research Center Program of the IBS (Institute for Basic Science) in Korea (grant no. IBS-R009-D1) and by the National Research Foundation of Korea (NRF) grant funded by the Korea government (MSIT) (No. 2018R1A5A6075964 and No. 2019R1C1C1002558). D.L. acknowledges the support by Samsung Electronics Co., Ltd. B.W. acknowledges the support by the NSF-MRSEC grant number DMR-1420620. The effort of L.-Q.C. is supported by National Science Foundation (NSF) through Grant No. DMR-1744213. The research at the University of Nebraska−Lincoln is supported by the National Science Foundation through the Nebraska Materials Research Science and Engineering Center (MRSEC), Grant No. DMR-1420645.

## Author contributions
D.L. conceived and designed the research. S.M.P. measured electrical transport under supervision of T.W.N. B.W. and L.-Q.C. carried out simulations of strain gradient and polarization. T.R.P., S.Y.P., L.T., and E.Y.T. carried out first-principles calculations. N.P. and D.S. prepared graphene samples. S.D. and J.R.K. fabricated thin films. E.K.K. and H.G.L. performed scanning electron microscopy imaging and estimation of tip radius. D.L. and S.M.P. wrote the paper with comments from all co-authors. D.L. directed the overall research.

## Competing interests
The authors declare no competing interests.
