## [Peer Review File · Nature Communications]

Reviewers' comments:

Reviewer #1 (Remarks to the Author):

The authors used atomic force microscopic tip to investigate ultrathin film of an archetypal dielectrics SrTiO₃ and discovered electrical state transition from insulating to conducting with 1e8 fold decrease of room temperature resistivity. The authors claimed that it is due to the flexoelectricity. More exactly, the tip induces strain gradient and thus finite polarization through flexoelectricity, and the induced polarization modifies the tunneling conductance of film. Nevertheless, the referee thinks that the reasoning and data, particularly the analysis of the strain profile, is insufficient to support their conclusion, and thus the referee does not recommend for publication.

Based on their contact mechanics model, the calculated strain gradient is around 1e7 m⁻¹. It is doubtful. The tip diameter is said be around 100nm, which is much larger than the thin film thickness of 3.9 nm with substrate. Considering these dimensions, the strain field in the region under the tip cannot really has large gradient.

Even if one can consider an ideal single centered force problem, which is more favorable to generate high gradient than the contact problem of round tip, one cannot arrive at such large strain gradient. Based on the Boussinesq's solution, the estimated strain gradient $\epsilon_{33,3}$ in the region near the surface (at the depth of 1nm) is only the region of 1e6 m⁻¹. The referee understands that there may be additional in-plane strain gradient due to the interface between the thin film and substrate. But it is hard to estimate the value from the presented results in Fig.2. Btw, the unit of the transverse strain is missing in the figure.

In the simulation model, periodic boundary conditions are assumed for the two in-plane direction. But this is not reasonable for a problem of force applied by the AFM tip. Moreover, contact mechanics model they used is a typical continuum model. But the grip spacing was set to be 0.1 nanometer, and the validity of a solution based on continuum model is high questionable.

In addition, the authors do not reason why the charge of AFM tip and the electrostatic of the thin film is not considered in the simulation. The polarization can in fact more dominantly be varied by electric field, since it is a first-order theory. Without excluding this fact, the conclusion made by the authors is hard to accept. Also, based on their tunneling current density model in relation to polarization, the referee wonders why the authors do not study a ferroelectric thin film, which has large polarization and permittivity.

Reviewer #2 (Remarks to the Author):

In the paper "colossal flexoresistance in dielectrics", Park et al. have claimed to realize colossal "flexoresistance" behavior in ultrathin SrTiO₃ (STO) films, which is very interesting. The authors exclude many factors that they believe will bring interference to the observed colossal flexoresistance behavior. However, it is far from clear that, in its present form, the authors can make a solid conclusion.

1. In a previous work (Ref.: Nat. Commun. 10, 537, 2019), the polarization induced local metallization of STO thin film has been reported by the same group. The main idea of the two papers is very similar. So, they should explain the similarities and differences between these two works in details.

2. The authors should pay attention to the flexoelectric coefficient value of STO that they set in the phase field simulations. Should the flexoelectric coefficient in STO thin film equals to that in their single crystal counterpart? In my view, once involving such a high strain gradient between the tip and thin film ($1E7 \text{ m}^{-1}$), the value of the average flexoelectric coefficient is questionable, since the induced flexoelectric polarization will be gradually saturated as the strain gradient increases.
3. The authors should also pay attention to the role of the surface charge compensation on the flexoresistance behavior. For the ultrathin STO thin film, the AFM tip motion at high pressures can simply mechanically remove screening charges and led to an unscreened surface. This will contribute significantly to the observed resistance reduction.
4. Much more important is that how to distinguish the triboelectric effect from the flexoelectric effect in this AFM scenario? Also, how to exclude the contributions from the surface piezoelectricity? (Refs.: Nature 538, 219-221, 2016; PRL 121, 057602, 2018)
5. The "flexoresistance" behavior is still questionable in other dielectrics. The authors find the same "electrical-state switching" in BTO thin film with a lower threshold loading force. However, it is known that the BTO ultrathin film ($<10 \text{ nm}$) deposited on SRO generally has a P-up (point from the substrate to the film) ferroelectric polarization state (Ref.: Adv. Mater. Interfaces 3, 1600737, 2016). This direction of ferroelectric polarization, which should be overcome, is opposite to the AFM tip-induced flexoelectric polarization. So, why the threshold loading force for BTO is lower than that of STO? Finally, to emphasize this concept, the flexoresistance behavior should be clearly demonstrated in other material systems.

Minor comments:

1. The author mentioned the growth of SRO as step flow mode in the text. However, the thickness of SRO has not been provided. As we know, the conductivity and the lattice parameters of SRO ultrathin films are strongly dependent on the thickness. The lattice will relax to its bulk value (3.93 \AA) when the film is thick enough.

Responses to the comments of Reviewer #1

We would like to thank the reviewer for reviewing our manuscript and providing critical points for us to consider, e.g., “Nevertheless, the referee thinks that the reasoning and data, particularly the analysis of the strain profile, is insufficient to support their conclusion”. We have read the reviewer’s comments very carefully and have undertaken further works to answer all of his/her comments. The main concern was with regard to the validity of the simulation results, especially the analysis of the strain profile. For clarity, we compare the strain distribution obtained numerically in our study with those calculated using two analytical contact mechanics models, that is, the Hertz model and Boussinesq’s solution. The Hertz model and Boussinesq’s solution consider the transversely isotropic elastic half-space in contact with a spherical indenter and with a point contact, respectively. The results are given below in Figure R1 and Table R1. We provide the reviewer’s comments followed by our detailed point-by-point responses (written in blue).

Question #1

Based on their contact mechanics model, the calculated strain gradient is around $1e7\text{ m}^{-1}$. It is doubtful. The tip diameter is said be around 100nm, which is much larger than the thin film thickness of 3.9 nm with substrate.

Response #1

As pointed out by the reviewer, the tip radius ($r_{\text{tip}} \sim 100\text{ nm}$) is much larger than the film thickness ($t \sim 4\text{ nm}$) in the model. However, please note that, when estimating strain gradients, we have to use the contact radius, not the tip radius itself. When the loading force is $15\text{ }\mu\text{N}$, the diamond tip of $r_{\text{tip}} \sim 100\text{ nm}$ contacts the SrTiO_3 surface with a contact radius of $a \sim 13\text{ nm}$ (calculated using the equations in Table 6 of Ref. 43 in the main text), comparable to the film thickness $t \sim 4\text{ nm}$. For the same reason, we adopted the closed-form solution of the stress distribution on the film surface in the “thin-film” limit (e.g., when $t \ll a$, based on the equations in Table 6 of Ref. 43 in the main text) as the boundary condition of the mechanical equilibrium equation in our phase-field model. To avoid further confusion, we stress the dimension of the contact radius in Line #84 of the revised manuscript as follows:

“Note that the actual contact radius is estimated to be around 13 nm for the case of a 15- μN tip loading force, which is much smaller than the tip radius r_{tip} .”

Question #2

Considering these dimensions, the strain field in the region under the tip cannot really has large gradient.

Response #2

In Responses #3 & #4, we show that, compared to the longitudinal (i.e., out-of-plane) strain u_{33} , the transverse (i.e., in-plane) strain u_{11} exhibits much larger variation across the film. As a result, the transverse strain gradient can be huge ($>10^7 \text{ m}^{-1}$) and then primarily contributes to the flexoelectric effect, consistent with findings from previous studies [see, e.g., “W. Chen *et al.*, *J. Mech. Phys. Solids* **111**, 43–66 (2018)”].

Question #3

Even if one can consider an ideal single concentrated force problem, which is more favorable to generate high gradient than the contact problem of round tip, one cannot arrive at such large strain gradient. Based on the Boussinesq’s solution, the estimated strain gradient $\epsilon_{33,3}$ in the region near the surface (at the depth of 1nm) is only the region of $1e6 \text{ m}^{-1}$.

Response #3

We thank the reviewer for suggesting the use of the classical Boussinesq’s solution to estimate the strain gradient. To test this possibility, we adopted the Boussinesq solution of an axisymmetric system for a transversely isotropic half-space elastic solid, to obtain the stress and strain field distributions subject to point force loading. To obtain the numerical values, material-related parameters for SrTiO₃ were used, such as Young’s modulus $E = 264 \text{ GPa}$ and Poisson’s ratio $\nu = 0.24$. The loading force was set to $15 \mu\text{N}$ for consistency with the conditions corresponding to Fig. 2 of the main text. The stress distribution expression for Boussinesq’s solution is given in classical books of contact mechanics (e.g., page 80 of “Fisher-Cripps, *Introduction to Contact Mechanics*”). Based on the stress distribution, we can calculate the strain distribution by assuming Hooke’s law for a pure elastic material. Furthermore, for comparison, we plot the strain distributions obtained by following the closed-form solution of the Hertz model with a spherical indenter (page 87 of the same book) and the numerical results obtained from our phase-field simulation. For the Hertz analytical model, we keep the same contact radius as in our numerical model, e.g., $a = 13.1 \text{ nm}$. The results are given below in Figure R1 and Table R1.

Figure R1 | Comparison of the strain distribution calculated using different models. a, Strain distributions calculated using Boussinesq’s solution (upper panel) and Hertz’s model of a spherical indenter (middle panel), and the numerical results of the present work (lower panel). u_{33} and u_{11} represent longitudinal (i.e., out-of-plane) and transverse (i.e., in-plane) strains, respectively. The contact center is at the upper-left corner of each plot. **b,c,** The strain profiles under the contact center as a function of the distance from the surface calculated using Boussinesq’s solution (**b**) and Hertz’s model (solid lines in **c**) in comparison with numerical results from the present work (dotted lines in **c**).

Table R1 | Comparison of strain gradients estimated using different models.

	Assumptions	Gradient of longitudinal strain ($\partial u_{33}/\partial x_3$) (m^{-1})	Gradient of transverse strain ($\partial u_{11}/\partial x_3$) (m^{-1})
Boussinesq’s solution	Half-space, transversely isotropic, nonpiezoelectric, point contact	-6.63×10^7 (at 10 nm in depth)	-1.95×10^7 (at 10 nm in depth)
Hertz’s solution	Half-space, transversely isotropic, nonpiezoelectric, spherical tip	4.59×10^6 (averaged over 0–4 nm in depth)	-1.18×10^7 (averaged over 0–4 nm in depth)
Present work	Thin film, anisotropic, piezo- and flexoelectric, spherical tip	5.16×10^6 (averaged over 0–3.5 nm in depth)	-1.55×10^7 (averaged over 0–3.5 nm in depth)

[*The sign of the strain gradient indicates whether the strain has intensified (+) or decayed (–) from the surface into the film/half space. For Boussinesq’s solution, the strain gradients estimated at a depth of 10 nm are shown, but when estimated at a depth of 1 nm, they become much larger ($\geq 10^8 \text{ m}^{-1}$).]

We can see that all three models (Boussinesq, Hertz, and the present numerical simulation) show huge strain gradients. In particular, the total transverse strain gradients (i.e., $\partial u_i/\partial x_3 = \partial u_{11}/\partial x_3 + \partial u_{22}/\partial x_3$) are as large as 3.9×10^7 , 2.4×10^7 , and $3.1 \times 10^7 \text{ m}^{-1}$ for the Boussinesq model, Hertz model, and present numerical simulations, respectively. Importantly, our numerical results compare very well with Hertz's analytical solution. There are some minor differences that may be attributed to the electromechanical interactions, including piezoelectric, ferroelectric, and flexoelectric effects, which were not considered in the analytical models.

Please also note that our results are consistent with those of many previous works [see, e.g., "Science **336**, 59 (2012)", "Science **360**, 904 (2018)", "J. Mech. Phys. Solids **111**, 43 (2018)", etc.], in showing a huge strain gradient ($>10^7 \text{ m}^{-1}$) in AFM tip-based experiments. Therefore, estimation of the strain gradient clearly shows that it can be as large as $>10^7 \text{ m}^{-1}$ near the surface, i.e., sufficiently large to cause strong flexoelectric effects.

Question #4

The referee understands that there may be additional in-plane strain gradient due to the interface between the thin film and substrate. But it is hard to estimate the value from the presented results in Fig.2

Response #4

We would like to thank the reviewer for suggesting the importance of the in-plane strain gradient. First, we must consider the gradient of transverse (i.e., in-plane) strain (e.g., $\partial u_{11}/\partial x_3$), because it could be much larger than the gradient of longitudinal (i.e., out-of-plane) strain (e.g., $\partial u_{33}/\partial x_3$). As shown in Table R1, both Hertz's model and our numerical simulation show that the total transverse strain gradient $\partial u_i/\partial x_3 (= \partial u_{11}/\partial x_3 + \partial u_{22}/\partial x_3)$ is one order of magnitude larger than the longitudinal strain gradient $\partial u_{33}/\partial x_3$. Thus, the transverse strain gradient primarily contributes to the flexoelectric effect, consistent with findings from previous studies [e.g., "W. Chen *et al.*, J. Mech. Phys. Solids **111**, 43–66 (2018)"]. This is why we focus on the transverse strain gradients $\partial u_i/\partial x_3$ in our manuscript.

Also, there exists the in-plane gradient of strains (e.g., $\partial u_{11}/\partial x_1$), inducing the in-plane polarization. However, please note that only the out-of-plane polarization should mainly contribute to the band crossing and resistivity change across the dielectric layer (Fig. 1 of the main text). Following the reviewer's comment, we revised Fig. 2b of the main text by labelling the contour lines. This revision would help the readers estimate strain gradients more easily.

Question #5

Btw, the unit of the transverse strain is missing in the figure.

Response #5

We thank the reviewer for pointing out this. Following the reviewer's comment, we added units (%) for the strain distributions in Fig. 2 of the main text.

Question #6

In the simulation model, periodic boundary conditions are assumed for the two in-plane direction. But this is not reasonable for a problem of force applied by the AFM tip. Moreover, contact mechanics model they used is a typical continuum model. But the grip spacing was set to be 0.1 nanometer, and the validity of a solution based on continuum model is high questionable.

Response #6

We thank the reviewer for pointing out this important issue. The reviewer expressed concern about the periodic boundary conditions imposed along the in-plane directions in our model. We believe that, as long as the lateral size is large enough to exclude interactions between the AFM tip with its neighboring periodical images, the 3D model with in-plane periodical boundary conditions is reasonable. In our simulation, we used $n_x = n_y = 128$ nm as the lateral dimension of the simulation box, which is large enough to exclude artificial interactions due to the periodic boundary conditions. We test this by plotting surface displacement under the tip contact along the surface across the tip contact center, as shown in Fig. R2 below. The surface displacement is reduced to a near-constant value at the two ends of the simulated system, which suggests that the AFM tip is "isolated" and has negligible interactions with its periodic images.

Figure R2 | Out-of-plane displacement on the SrTiO₃ surface under a 15- μ N mechanical force calculated by a spherical indenter.

The reviewer also questions the applicability of a continuum-based contact mechanics model to the nanoelectromechanics problem of an AFM tip. However, continuum-based contact mechanics have been used extensively to model AFM, PFM, nanoindentation, and other contact mechanics problems at the nanoscale in solids. We agree that the atomistic structure may be important when the surface roughness is at the atomistic scale, so that atomistic approaches such as molecular dynamics would become necessary [e.g., “Y. Mo *et al.*, Friction laws at the nanoscale. *Nature* **457**, 1116–1119 (2009)”.]. However, this is not the case in the present study, because the as-grown SrTiO₃ film surface possesses near-perfect flatness (as shown in Fig. 4a of the main text) through the control exerted in our experiments.

The reviewer also questioned our choice of grid size. When numerically solving partial differential equations for mechanics problems, a smaller grid size generally gives rise to more accurate solutions, although at the expense of computational resources. We used a size of 0.1 nm along the out-of-plane direction of the film, to ensure sufficient resolution in this direction and accuracy of the numerical results. Thus, the choice of grid size is not relevant to the validity of the continuum model.

Question #7

In addition, the authors do not reason why the charge of AFM tip and the electrostatic of the thin film is not considered in the simulation. The polarization can in fact more dominantly be varied by electric field, since it is a first-order theory. Without excluding this fact, the conclusion made by the authors is hard to accept.

Response #7

We thank the reviewer for pointing out this crucial issue. The reviewer questions the absence of the electrostatic interactions between the AFM tip and the thin film surface. He/she believes that SrTiO₃ can become polarized due to the electrostatic field, e.g., generated by the charge in the AFM tip. However, if the polarization in SrTiO₃ is caused by the charge of the AFM tip, our main observation (i.e., colossal resistivity change) would be almost independent of the tip radius in the experiments, because the charge density on the tip would not vary much with the tip radius. In contrast, we do observe a strong dependence of the resistivity change on the AFM tip radius (Fig. 3 of the main text). This result suggests that the strain gradient, which becomes much smaller with a larger AFM tip radius, is the primary cause of the observed resistivity change.

Furthermore, we carried out additional experiments for estimating the electrostatic field (e.g., built by the charge in the AFM tip) in our experimental geometry. If there exists a built-in electrostatic field, then we can detect it from the shift in the polarization-switching hysteresis loops of BaTiO₃. As shown in Fig. R3 below, we observe that in 10 unit cell-thick (i.e., ~4 nm-thick) BaTiO₃, the hysteresis loops (measured with a sufficiently low loading force) are shifted by around 0.1 V. Accordingly, the built-in electrostatic field is measured as, at most, $3 \times 10^7 \text{ V m}^{-1}$ for our experimental geometry. However, the colossal decrease in resistivity requires a much larger threshold electric field $E_{\text{th}} \sim \frac{\Delta_{\text{bg}}}{e \cdot t} \sim 8 \times 10^8 \text{ V m}^{-1}$ [Eq. (3) of the main text], where Δ_{bg} and t represent the bandgap and thickness of the dielectric layer, respectively. Furthermore, given the dielectric permittivity $\epsilon \sim 20\epsilon_0$ of strained SrTiO₃ and BaTiO₃ (Supplementary Fig. 7), the electrostatic field of $3 \times 10^7 \text{ V m}^{-1}$ can generate an electric polarization of up to 0.005 C m^{-2} , which is much smaller than the flexoelectric polarization of around 0.2 C m^{-2} .

Figure R3 | Piezoresponse force microscopy (PFM) studies. a,b, PFM phase (a) and amplitude (b) hysteresis loops of 10 unit cell-thick BaTiO₃. We measure PFM hysteresis loops using a sufficiently low loading force.

Based on the above, we believe that any electrostatic interactions (e.g., related to the charge of the AFM tip) can be safely excluded in our model. This result was added to the revised Supplementary Information as Supplementary Fig. 11.

Question #8

Also, based on their tunneling current density model in relation to polarization, the referee wonders why the authors do not study a ferroelectric thin film, which has large polarization and permittivity.

Response #8

We would like to thank the reviewer for pointing out the possibility of using ferroelectric materials. Bulk ferroelectric materials could have a large polarization. However, in the ultrathin limit, the value of ferroelectric polarization would be highly suppressed compared to the bulk value due to intrinsic size effects [see, e.g., “J. Junquera & P. Ghosez, *Nature* **422**, 506–509 (2003)”. For example, a previous experimental work revealed that the remnant polarization of ferroelectric BaTiO₃ becomes suppressed down to $\sim 0.01 \text{ C m}^{-2}$ in the ultrathin limit [see, e.g., “H. Lu *et al.*, *Adv. Mater.* **24**, 1209–1216 (2012)”. This is much smaller than the flexoelectric polarization (0.2 C m^{-2} on average) induced by a diamond tip with a loading force of $15 \mu\text{N}$. Therefore, in the AFM tip-based experiment, a large polarization in ultrathin dielectrics can be generated most effectively via flexoelectricity.

Responses to the comments of Reviewer #2

We would like to thank the reviewer for his/her in-depth review and excellent questions/suggestions regarding our manuscript. We have read the reviewer's comments very carefully and have undertaken further works to answer all of his/her questions. In the pages that follow, we provide our responses to each of the reviewer's comments, in order. The responses are written in blue.

In the paper "colossal flexoresistance in dielectrics", Park et al. have claimed to realize colossal "flexoresistance" behavior in ultrathin SrTiO₃ (STO) films, which is very interesting.

We thank the reviewer for stating that our work is very interesting.

Question #1

In a previous work (Ref.: Nat. Commun. 10, 537, 2019), the polarization induced local metallization of STO thin film has been reported by the same group. The main idea of the two papers is very similar. So, they should explain the similarities and differences between these two works in details.

Response #1

Regarding the similarity between the papers, we utilized the depolarization field induced by flexoelectric polarization to modify the tunnel barrier profile of an ultrathin dielectric. Also, for generating a flexoelectric polarization, we adopted the AFM tip-based experimental method in both the previous and current works.

However, the current work is distinctly different from the previous work, in terms of both its purpose and achievement. Our previous work [Nat. Commun. **10**, 537 (2019)] focused mainly on how to estimate the flexocoupling coefficient at the nanoscale under high strain gradients. Then, we successfully demonstrated an effective way for characterizing nanoscale flexoelectricity under high strain gradients. Importantly, in our previous work, we mainly considered the regime of flexoelectric polarization, where a dielectric layer remains insulating. On the other hand, the current work focuses on how to achieve static, damage-free control of electrical states in dielectrics, which has remained a great challenge. To do this, we considered the regime of larger flexoelectric polarization, where the conduction and valence bands of

SrTiO₃ could cross each other. As discussed in our manuscript, under these conditions, the “whole” SrTiO₃ layer behaves as a conductor, due to the highly enhanced tunnel conductance and/or Zener breakdown. Therefore, for the first time, this work demonstrated static, damage-free control of electrical states in an otherwise highly insulating dielectric. This represents a fundamental breakthrough that provides new insight into, and thus could improve, the electrical control of solids.

To conclude, the purpose and achievement of the current work are obviously different from those of the previous work. We have added text pertaining to the difference between the previous and current work to the revised Supplementary Information, as Supplementary Note 1. We would like to thank the reviewer again for pointing out this issue and believe that our revision has improved the clarity of our work.

Question #2

The authors should pay attention to the flexoelectric coefficient value of STO that they set in the phase field simulations. Should the flexoelectric coefficient in STO thin film equals to that in their single crystal counterpart? In my view, once involving such a high strain gradient between the tip and thin film ($1E7$ m⁻¹), the value of the average flexoelectric coefficient is questionable, since the induced flexoelectric polarization will be gradually saturated as the strain gradient increases.

Response #2

We completely agree with the reviewer’s point that, under high strain gradients, the flexocoupling coefficient of SrTiO₃ thin film could be different from that of SrTiO₃ bulk. In fact, this is the reason why we first measured the effective flexocoupling coefficient of SrTiO₃ under high strain gradients in our previous work [Nat. Commun. **10**, 537 (2019)]. Interestingly, in our previous work, we discovered the enhanced flexocoupling coefficient of SrTiO₃ under high strain gradients, and attributed it to a nonlinear flexoelectric response and/or a surface contribution. Therefore, as mentioned in Line #320 of the manuscript, our phase field simulation adopted the value of the flexocoupling coefficient measured in our previous work.

Question #3

The authors should also pay attention to the role of the surface charge compensation on the flexoresistance behavior. For the ultrathin STO thin film, the AFM tip motion at high pressures

can simply mechanically remove screening charges and led to an unscreened surface. This will contribute significantly to the observed resistance reduction.

Response #3

We thank the reviewer for bringing up the surface (screening) charge issue. First, we would like to point out that our on-off experiments (Fig. 4c) and experiment with a graphene electrode (Supplementary Fig. S10) can overcome the screening charge removal issue. If the resistance reduction is due to the screening charge removal induced by mechanical loading, the response cannot be reversible in an on-off test.

Furthermore, based on simple electrostatics [Phys. Rev. Lett. **94**, 246802 (2005)], we find that the screening charge density σ_s tends to zero in the ultrathin limit (as in our case):

$$\sigma_s = \frac{Pd}{\varepsilon(\delta_1 + \delta_2) + d},$$

where P , d , and ε indicate polarization, thickness, and the dielectric constant of the dielectric layer, respectively. δ_1 and δ_2 are the Thomas–Fermi screening lengths in the electrodes. Therefore, the absolute value of σ_s should change negligibly by any means, as it is already saturated to almost zero.

Question #4

Much more important is that how to distinguish the triboelectric effect from the flexoelectric effect in this AFM scenario?

Response #4

We would like to thank the reviewer for pointing out this crucial issue. If the observed resistance change originates from the triboelectric effect, it would be larger when using the AFM tip with a larger tip radius, where the contact area between the tip and sample becomes larger. The triboelectric effect is related to the degree of electrical charge transfer between the materials; thus, the larger the contact area, the higher the change in the charge transfer. However, our experimental observation is in opposition to this expectation. As the tip radius increased, the resistivity change was suppressed significantly. This excludes the triboelectric effect as the primary origin of the colossal resistivity change observed.

Furthermore, using Kelvin probe force microscopy (KPFM), we measured the surface potential (directly related to the surface charge state) before and after mechanical loading. As shown in

Fig. R4 below, the KPFM experiments showed almost no change in the surface potential, indicating that the mechanical loading in our experiments did not induce or change the surface charge. Thus, the triboelectric effect cannot be the primary reason for the colossal resistivity change observed. This result was added to the revised Supplementary Information as Supplementary Fig. 12.

Figure R4 | Kelvin probe force microscopy (KPFM) studies. **a**, KPFM image showing surface potential profiles of SrTiO₃, after applying AFM tip loading force (~15 μN). Regions where the mechanical loadings were applied are marked by yellow circles. **b**, Line profiles of the surface potential, measured along the paths A, B, and C, denoted by dashed lines in **a**.

Question #5

Also, how to exclude the contributions from the surface piezoelectricity? (Refs.: Nature 538, 219-221, 2016; PRL 121, 057602, 2018)]

Response #5

We thank the reviewer for asking about the contribution of the surface piezoelectricity. We did not exclude surface piezoelectricity; in fact, we cannot exclude it. Surface piezoelectricity has been regarded as an actual intrinsic component of the total flexoelectric response [see, e.g., “Nature 538, 219 (2016)”, “Nanotechnology 24, 432001 (2013)”, “Phys. Rev. Lett. 115,

037601 (2015)”, “Phys. Rev. B **90**, 201112(R) (2014)”. That is, the surface contribution, combined with the bulk contribution, determines the total flexocoupling coefficient. Thus, this study used the total flexocoupling coefficient, measured in our previous study [Nat. Commun. **10**, 537 (2019)]. Note that in our previous work, we suggested that the total flexoelectric response in ultrathin dielectrics could be enhanced due to the surface contribution.

Question #6

The “flexoresistance” behavior is still questionable in other dielectrics. The authors find the same “electrical-state switching” in BTO thin film with a lower threshold loading force. However, it is known that the BTO ultrathin film (<10 nm) deposited on SRO generally has a P-up (point from the substrate to the film) ferroelectric polarization state (Ref.: Adv. Mater. Interfaces **3**, 1600737, 2016). This direction of ferroelectric polarization, which should be overcome, is opposite to the AFM tip-induced flexoelectric polarization. So, why the threshold loading force for BTO is lower than that of STO?

Response #6

We thank the reviewer for pointing out this important issue. We have performed PFM experiments to measure the self-polarization of as-grown BaTiO₃. Please note that the direction of self-polarization in as-grown BaTiO₃ depends on many parameters, such as deposition temperature, pressure, annealing conditions, strain state, etc. As shown in Fig. R5 below, the self-polarization in our BaTiO₃ film is directed downward. Thus, the ferroelectric polarization in BaTiO₃ could lower the threshold loading force in our experiments. This result was added to the revised Supplementary Information as Supplementary Fig. 11.

Figure R5 | Piezoresponse force microscopy (PFM) studies of BaTiO₃/SrRuO₃/SrTiO₃ heterostructure. a, Surface topography image of 10 unit cell-thick BaTiO₃. **b**, PFM phase images after electric bias applied. Black and yellow regions correspond to the region with upward and downward polarizations, respectively.

Furthermore, our experimental geometry induces compressive strain in both the transverse and longitudinal directions, attributable to AFM tip-induced downward bending and pressing. Such three-dimensional compression could weaken ferroelectricity in a dielectric layer (e.g., BaTiO₃). Therefore, we do not actually have to be concerned with the ferroelectric polarization of BaTiO₃. Instead, as mentioned in our manuscript, the flexocoupling strength of BaTiO₃ could be inherently larger than that of SrTiO₃ [J. Narvaez *et al.*, *Phys. Rev. Lett.* **115**, 037601 (2015)], leading to the lower threshold loading force for BaTiO₃.

Question #7

Finally, to emphasize this concept, the flexoresistance behavior should be clearly demonstrated in other material systems.

Response #7

We would like to thank the reviewer for bringing up the universality issue. We have tested the “flexoresistance” behavior in other dielectric materials, i.e., CaTiO₃, as shown in Fig. R6 below. A 10 unit cell-thick CaTiO₃ film was grown on a LaAlO₃ substrate buffered by a LaNiO₃ conducting electrode. The measured I - V results for incremental loading forces confirmed a linear relationship for loading forces exceeding 18 μ N. This threshold loading force (18 μ N) for CaTiO₃ is slightly higher compared to those of SrTiO₃ (15 μ N) and BaTiO₃ (12 μ N). This could be due to the larger band gap (3.8 eV) of CaTiO₃ compared to those of SrTiO₃ (3.2 eV) and BaTiO₃ (3.2 eV). In addition, we also tested the on-off behavior; indeed, the behavior was reversible (Fig. R6). This result was added to the revised Supplementary Information as Supplementary Fig. 14.

Figure R6 | Reversible flexoresistance behavior in a 10 unit cell-thick CaTiO₃ film. Current measured with a 0.1-V bias voltage under two representative loading forces in CaTiO₃. During the measurements, we set the current limit (compliance) to 20 nA.

Question #8

The author mentioned the growth of SRO as step flow mode in the text. However, the thickness of SRO has not been provided. As we know, the conductivity and the lattice parameters of SRO ultrathin films are strongly dependent on the thickness. The lattice will relax to its bulk value (3.93 Å) when the film is thick enough.

Response #8

We thank the reviewer for pointing out this issue. We have performed an XRD reciprocal space map for the SrTiO₃/SrRuO₃/SrTiO₃ heterostructure to confirm the fully strained state of the SrRuO₃ layer, as shown in Fig. R7 below. This result was added to the revised Supplementary Information as Supplementary Fig. 5. Also, we added the thickness (i.e., 20 nm) of the SrRuO₃ layer to Line #261 of the revised manuscript.

Figure R7. Structural characterizations of STO thin films. **a**, X-ray diffraction (XRD) 2θ - ω scan of an STO thin film grown on a (001)-oriented STO substrate, with a conductive SrRuO₃ (SRO) buffer layer. The diffraction peak of SRO is indexed in pseudocubic perovskite notation. **b**, XRD reciprocal space mapping measured from the STO/SRO/STO (001) film around (103) diffraction. The red dashed line indicates that SRO and STO thin films are fully strained on the STO (001) substrate. SRO (103) and STO (103) diffractions share the same in-plane lattice constant as 3.905 Å.

Reviewers' comments:

Reviewer #1 (Remarks to the Author):

The authors have made large effort to reason their conclusion of flexoresistance. The referee appreciate it greatly. Nevertheless, the referee is still concerned with other possible reasons for the observed change in the resistance.

For instance, one is the electrostatic interactions between the AFM tip and the thin film surface. The authors argued that "if the polarization in SrTiO₃ is caused by the charge of the AFM tip, our main observation (i.e., colossal resistivity change) would be almost independent of the tip radius in the experiments, because the charge density on the tip would not vary much with the tip radius." However, even if the charge density on the tip does not change, the size variation of the tip still can induce changes in the electric response of the thin film, if the tip size is comparable to the size of the gradient field. It is indeed the case, according to the reported data (the contact area radius around 13nm, the gradient field region is around 30nm).

Moreover, regarding the ferroelectric dielectrics, the answers of the authors to the two referees seem inconsistent. On one hand, the authors argue that "in the ultrathin limit, the value of ferroelectric polarization would be highly suppressed compared to the bulk value due to intrinsic size effects", therefore the use of ferroelectric materials is not favored in their study of flexoresistance. (By the way, in their reply to the comment #8 of referee #1, they argued only w.r.t the polarization, but no comment on the higher permittivities of ferroelectrics, which in fact should also be the reason for larger flexocoupling strength of BaTiO₃). On the other hand, in the reply to comment #6 of the referee #2 regarding "why the threshold loading force for BTO is lower than that of STO"? The authors seemingly try very hard to convince that the BTO is more favorable to have flexoresistance than STO. It is quite confusing.

Reviewer #2 (Remarks to the Author):

It's my pleasure to find all my comments have been well clarified. Excellent work!

Responses to the comments of Reviewer #1

We would like to thank the reviewer for reviewing our manuscript and greatly appreciate his/her high evaluation of our efforts, e.g., “The authors have made large effort to reason their conclusion of flexoresistance. The referee appreciate it greatly.” Also, we would like to thank the reviewer for providing other points to consider. We have read and responded to all of the reviewer’s comments very carefully. In the pages that follow, we provide our responses to each of the reviewer’s comments, in order. The responses are written in blue.

Question #1

For instance, one is the electrostatic interactions between the AFM tips and the thin film surface. The authors argued that “if the polarization in SrTiO₃ is caused by the charge of the AFM tip, our main observation (i.e., colossal resistivity change) would be almost independent of the tip radius in the experiments, because the charge density on the tip would not vary much with the tip radius.” However, even if the charge density on the tip does not change, the size variation of the tip still can induce changes in the electric response of the thin film, if the tip size is comparable to the size of the gradient field. It is indeed the case, according to the reported data (the contact area radius around 13nm, the gradient field region is around 30nm).

Response #1

We thank the reviewer for suggesting that even with the same charge density, AFM tips with different tip radii (r_{tip}) could elicit different electric responses from the thin film. The reviewer seems to believe that this may explain the observed r_{tip} -dependence of the resistance change. To assess this possibility, we approximately estimated the electric field generated by a charged AFM tip. Here, we simplify the situation, as shown below in Fig. R1, and then consider the electric field generated by a uniformly charged circular disk with radius R .

[Fig. R1: Simplified experimental geometry. R corresponds to the contact radius. σ is the surface charge density of the conducting tip.]

For this finite-sized charged disk, the out-of-plane electric field will decrease in magnitude as R decreases. [Please note that only the out-of-plane electric field (and polarization) should mainly contribute to the band crossing and resistance change across the dielectric layer (Fig. 1 of the main text).] For instance, a simple electrostatic calculation yields the out-of-plane electric field E_{OOP} at a distance z from the center [i.e., dashed line in Fig. R1(b)], as follows:

$$E_{\text{OOP}}(z) = \frac{\sigma}{\varepsilon} \left[1 - \frac{z}{(z^2 + R^2)^{1/2}} \right],$$

where σ is the surface charge density of the circular disk and ε is the dielectric permittivity of the dielectric layer. Since the contact radius R decreases with decreasing r_{tip} , the AFM tip with a smaller r_{tip} should induce a smaller average E_{OOP} (as shown in Fig. R2 below), thereby reducing the resistance change. However, this is opposite to our experimental observations. We observed colossal resistance change only in the case using a sharp AFM tip (with a small r_{tip}). In addition to this opposite trend, as R varies within the experimentally achievable range, the average E_{OOP} changes by only <10% (Fig. R2), which is too small to be attributable to the colossal resistance change and its r_{tip} -dependence.

[Fig. R2: With decreasing R , $E_{\text{OOP}}(z)$, induced by σ , decreases on average. Here, $E_{\text{OOP}}(z)$ is calculated along the center line.]

Therefore, the electrostatic interactions between the AFM tip and thin film cannot explain the observed r_{tip} -dependence of the resistance change. Furthermore, importantly, as already demonstrated experimentally in our previous response, such electrostatic interactions can only generate a negligible electric field in our experimental geometry, much smaller than that generated by flexoelectricity. Considering these qualitative and quantitative aspects, we exclude the electrostatic interaction between the AFM tip and the thin film as the primary origin of the observed resistance change.

Question #2

Moreover, regarding the ferroelectric dielectrics, the answers of the authors to the two referees seem inconsistent. On one hand, the authors argue that “in the ultrathin limit, the value of ferroelectric polarization would be highly suppressed compared to the bulk value due to intrinsic size effects”, therefore the use of ferroelectric materials is not favored in their study of flexoresistance. (By the way, in their reply to the comment #8 of referee #1, they argued only w.r.t the polarization, but no comment on the higher permittivities of ferroelectrics, which in fact should also be the reason for larger flexocoupling strength of BaTiO₃). On the other hand, in the reply to comment #6 of the referee #2 regarding “why the threshold loading force for BTO is lower than that of STO”? The authors seemingly try very hard to convince that the BTO is more favorable to have flexoresistance than STO. It is quite confusing.

Response #2

We thank the reviewer for this comment, but we believe that there are some misunderstandings. In particular, we did not insist that the use of ferroelectric materials is not favored in our study of flexoresistance. Our point is that the contribution of ferroelectric polarization itself would not be so important to our experimental observations; instead, the contribution of flexoelectric polarization is the main reason for the observed resistance change. This means that some of the conventional ferroelectric materials (e.g., BTO) could be favorable for our flexoresistance study, as long as their flexocoupling coefficient is large. Therefore, we do not believe that our responses are inconsistent. A more detailed argument follows below.

Equations (2) and (3) in the main text approximate the threshold strain gradient $(\partial u/\partial x)_{\text{th}}$ as

$$\left(\frac{\partial u}{\partial x}\right)_{\text{th}} = \left(\frac{1}{f_{\text{eff}}}\right) \cdot \frac{\Delta_{\text{bg}}}{e \cdot t},$$

where f_{eff} is the effective flexocoupling coefficient, e is the electronic charge, and Δ_{bg} and t are the bandgap and thickness of the dielectric layer, respectively. Thus, the lower threshold loading force for BTO originates from its larger f_{eff} compared to that of STO. Indeed, it was experimentally observed that BTO inherently could have a much larger f_{eff} than STO, regardless of its ferroelectric and paraelectric phases [J. Narvaez *et al.*, *Phys. Rev. Lett.* **115**, 037601 (2015)]. Importantly, the **flexocoupling** coefficient f_{eff} (in the unit of V) is

independent of the dielectric permittivity ε of the materials, and is therefore a more fundamental quantity [see, e.g., “M. Stengel, Phys. Rev. B **90**, 201112(R) (2014)”. Instead, the flexoelectric coefficient μ_{eff} (in the unit of C/m) linearly scales with ε , following the relation $\mu_{\text{eff}} = \varepsilon \cdot f_{\text{eff}}$.

Responses to the comments of Reviewer #2

We would like to thank the reviewer for carefully reviewing our manuscript. In addition, we greatly appreciate his/her high evaluation of our work, e.g., “It’s my pleasure to find all my comments have been well clarified. Excellent work!” Owing to the invaluable comments of the reviewer, we were able to improve the overall quality of our manuscript significantly.

REVIEWERS' COMMENTS:

Reviewer #1 (Remarks to the Author):

Thanks the authors for explaining their misleading reply to my comments on ferroelectrics and for confirming my suggestion that some conventional ferroelectric materials (e.g., BTO) could be favorable for our flexoresistance study, as long as their flexocoupling coefficient is large.

As for the electrostatic influences, the referee keeps the opinion. Since it is only one possible factor, it may not necessarily prevent the publication acceptance of this manuscript.

Responses to the comments of Reviewer #1

We would like to thank the reviewer for reviewing our manuscript and really appreciate his/her high evaluation of our effort, e.g., “Thanks the authors for explaining their misleading reply to my comments on ferroelectrics and for confirming my suggestion that some conventional ferroelectric materials (e.g., BTO) could be favorable for our flexoresistance study, as long as their flexocoupling coefficient is large.” Also, we would like to thank the reviewer for recommending our paper to be published in Nature Communications, e.g., “Since it is only possible factor, it may not necessarily prevent the acceptance of this manuscript.”

Question #1

As for the electrostatic influences, the referee keeps the opinion.

Response #1

As we have already explained in the last revision, the electrostatic interactions between the AFM tips and the thin film surface cannot explain the observed r_{tip} -dependence of the resistance change. Furthermore, importantly, as already demonstrated experimentally in our previous response, such electrostatic interactions can only generate a negligible electric field in our experimental geometry, much smaller than that by flexoelectricity. Therefore, based on these evidences, we strongly believe that the electrostatic influences cannot explain the observed colossal resistance change.

In our revised manuscript, we clarified this by stating “In addition, based on quantitative and qualitative evidences (Supplementary Figs. 11–13), we exclude an electrostatic interaction between the AFM tip and STO as the primary origin of the colossal resistivity change observed.”